# A WALK WITH SGD: HOW SGD EXPLORES REGIONS OF DEEP NETWORK LOSS?

## ABSTRACT

The non-convex nature of the loss landscape of deep neural networks (DNN) lends them the intuition that over the course of training, stochastic optimization algorithms explore different regions of the loss surface by entering and escaping many local minima due to the noise induced by mini-batches. But is this really the case? This question couples the geometry of the DNN loss landscape with how stochastic optimization algorithms like SGD interact with it during training. Answering this question may help us qualitatively understand the dynamics of deep neural network optimization. We show evidence through qualitative and quantitative experiments that mini-batch SGD rarely *crosses* barriers during DNN optimization. As we show, the mini-batch induced noise helps SGD explore different regions of the loss surface using a seemingly different mechanism. To complement this finding, we also investigate the qualitative reason behind the slowing down of this exploration when using larger batch-sizes. We show this happens because gradients from larger batch-sizes align more with the top eigenvectors of the Hessian, which makes SGD oscillate in the proximity of the parameter initialization, thus preventing exploration.

## 1 INTRODUCTION

The non-convexity of the deep neural network (DNN) loss surface makes the behavior of optimization algorithms less intuitive compared to the convex setting. Moreover, optimization in DNNs is no longer about finding any minimum, but rather about finding ones that generalizes well (Keskar et al., 2016). Since deep networks are initialized randomly, finding such minima will require exploration of different regions of the loss surface. This intuition has been formalized in recent papers that study stochastic gradient descent (SGD) as a diffusion process (Hoffer et al., 2017; Smith & Le, 2017; Jastrzebski et al., 2017; Chaudhari & Soatto, 2017). Briefly, these papers show that SGD simulates a discrete approximation of stochastic differential equation (SDE), and hence performs a random walk on the potential induced by the DNN loss surface.

In this work, we complement the diffusion perspective of SGD with a qualitative view of how SGD explores different regions of the non-convex loss landscape of deep neural networks through empirical evidence. Intuitively, when performing random walk on a potential, one would expect barriers being crossed quite often during the process. We show in this work that SGD rarely *crosses* any barriers along its path during the course of training. By this observation, we do not claim that SGD does not simulate diffusion. Through experimental deductions, we show an alternate mechanism that SGD seems to dominantly use to explore different regions of the non-convex loss landscape.

Further, it is known that larger batch-sizes slow down the diffusion process (Hoffer et al., 2017). We show the qualitative reason behind this slow down to be an oscillation behavior of SGD which prevents it from moving far away from initialization. This behavior is a result of the mini-batch gradients becoming increasingly aligned with the top eigenvectors of the Hessian for larger batch-sizes. This behavior is known to slow down convergence in optimization theory (for instance consider the motivation behind momentum (Polyak, 1964; Sutskever et al., 2013)). We discuss how it also slows down explorations in the non-convex setting of deep network loss surface.

Experiments are conducted on multiple data sets, architectures and hyper-parameter settings. The findings mentioned above hold true on all of them.

## 2 Setup

We now describe the details of how we study the existence of barriers along the optimization path of SGD. The main tool we use for studying the DNN loss surface along SGD's path is to interpolate the loss surface between parameters before and after each training update. We note that this strategy of interpolating the loss surface between parameters was introduced by Goodfellow et al. (2014). In their paper, the interpolation is conducted between initial and final (after training) parameter values for analysis purposes. In contrast, we compute interpolations before and after each training update because this interpolation precisely tells us whether or not SGD crosses a barrier during an update step. We say a *barrier is crossed* when we see a point in the parameter space interpolated between the parameters just before and after an update step, such that the loss at the barrier point is higher than the loss at both the other points.

Consider that the parameters $\theta$ of a neural network are initialized to a value $\theta_0$. When using an optimization method to update these parameters, the $t^{th}$ update step takes the parameter from $\theta_t$ to $\theta_{t+1}$ using estimated gradient $\mathbf{g}_t$ as,

$$\theta_{t+1} = \theta_t - \eta \mathbf{g}_t \tag{1}$$

where $\eta$ is the learning rate. Notice the $t^{th}$ update step implies the $t^{th}$ epoch *only* in the case when using the full batch gradient descent (GD). In the case of stochastic gradient descent, one iteration is an update from gradient computed from a mini-batch. We then interpolate the DNN loss between the convex combination of $\theta_t$ and $\theta_{t+1}$ by considering parameter vectors $\theta_t^\alpha = (1-\alpha)\theta_t + \alpha\theta_{t+1}$, where $\alpha \in [0,1]$ is chosen such that we obtain 10 samples uniformly placed between these two parameter points. We note that even though the updates are performed using mini-batches for SGD, the training loss values we compute for the interpolation use the full dataset to visualize the actual loss landscape.

## 3 Barriers and Exploration during SGD Training

For this section, we perform experiments on MNIST (Lecun & Cortes) and CIFAR-10 (Krizhevsky, 2009) datasets, and use multi-layer perceptrons (MLP), VGG-11 (Simonyan & Zisserman, 2014) and Resnet-56 (He et al., 2016) architectures with various batch sizes and learning rates. We discuss our observations for VGG-11 architecture on CIFAR-10 dataset (figure 1) as a reference but the same conclusions hold for experiments on MLP trained on MNIST (figure 3) and Resnet-56 trained on CIFAR-10 (figure 2).

We train VGG-11 on CIFAR-10 with a batch size of 100 and fixed learning rate of 0.1. We report the visualization of loss interpolation between consecutive iterations for 40 iterations from epochs 1, 2, 25 and 100 for visual clarity. The interpolation is shown in figure 1. To be clear, the x-axis is calibrated by the number of iterations, and there are 10 interpolated loss values between each consecutive iteration (vertical gray lines) in the *training loss* plots. In these plots, we find two interesting behavior of SGD.

First, we find that the interpolated loss between every consecutive update from SGD optimization update appears to be a quadratic-like structure with a minimum in between. Note that while this is not visible for epochs 25 and 100, we later show quantitative measurements that ensures this claim. This plot thus shows that in the iterations plotted, SGD rarely *crosses* barriers.

Second, we observe how the minimum of each interpolation evolves as training progresses. This is highlighted in figure 1 (a) with a dashed orange line. We find that this minimum has ups and downs along the path of SGD for all our interpolation plots. To draw deductions from this observation, consider a simple example that helps us understand this scenario concretely. Let parameter points $\theta_A$, $\theta_B$ and $\theta_C$ be a result of three consecutive SGD updates with loss values $\ell_A$, $\ell_B$ and $\ell_C$ (using full training set). Note that since these are only three points, they exist in a two dimensional subspace and the loss value can be imagined along the third dimension. Then corresponding to the behavior in the plot, there is a parameter point $\theta_{AB}$ between $\theta_A$ and $\theta_B$ on the line connecting these two points, which has a loss value $\ell_{AB} < \ell_A, \ell_B$. Similarly there is a point $\theta_{BC}$ between $\theta_B$ and $\theta_C$ on the line connecting these two points, which has a loss value $\ell_{BC} < \ell_B, \ell_C$. Given this construction, for any configuration of $\theta_A$, $\theta_B$ and $\theta_C$ on the two dimensional plane, it is easy to see that if $\ell_{AB} < \ell_{BC}$, any path from $\theta_{AB}$ to $\theta_{BC}$ will have loss values that must increase at some point. Hence, what

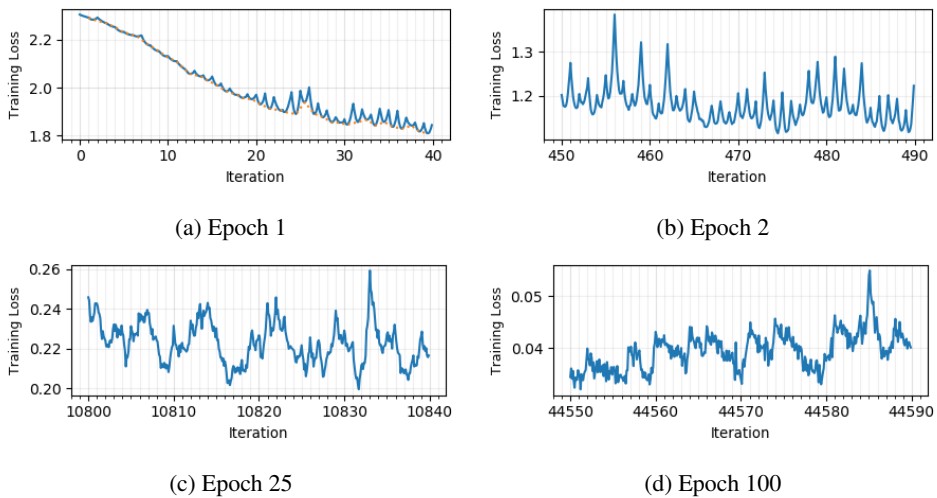

Figure 1: Plots for VGG-11 architecture trained using SGD on CIFAR-10. Each plot contains the training loss for 40 iterations of training at various epochs. Between the training loss at every consecutive iteration (vertical gray lines), we uniformly sample 10 points between the parameters before and after the training update and calculate the loss at these points. Thus we take a slice of the loss surface between two iterations. These loss values are plotted between every consecutive training loss value from training updates. We find that the loss interpolation between consecutive iterations have a minimum in between in all cases showing barriers are not being *crossed*. For epochs 25 and 100 this is not clearly visible, but we quantitatively record it and discuss it later. The dashed orange line (only shown in the epoch 1 plot) connects the minimum of the loss interpolation between consecutive iterations and is shown to highlight that the valley floor has ups and downs along the path of SGD (which can be seen for all epochs).

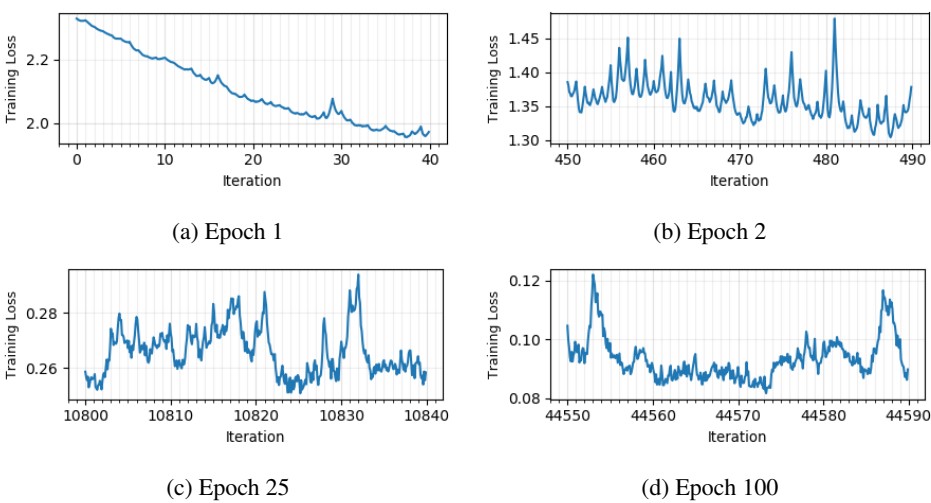

Figure 2: Plots for Resnet-56 architecture trained using SGD on CIFAR-10. All the descriptions are same as described in figure 1.

this construction essentially represents (as we refer to it), is a situation where SGD has moved *over* a barrier. Therefore, the ups and downs of the minimum between loss interpolations in figure 1 (a,b,c,d) represents SGD moving over barriers. In this way we find that when running SGD on the loss surface of deep networks, instead of crossing barriers, a more dominant way SGD performs exploration is by moving over them.

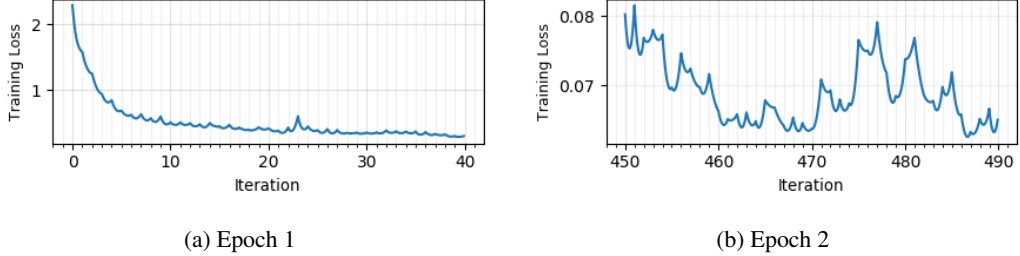

(a) Epoch 1          (b) Epoch 2

Figure 3: Plots for MLP architecture trained using SGD on MNIST. All the descriptions are same as described in figure 1.

| Arch\Epochs | 1 | 10 | 25 | 100 |
|---|---|---|---|---|
| **VGG-11** | 0 | 0 | 5 | 13 |
| **Resnet-56** | 0 | 0 | 2 | 23 |
| **MLP** | 0 | 3 | 5 | - |

Table 1: Number of barriers crossed during training of one epoch (450 iterations) for VGG-11 and Resnet-56 on CIFAR-10 and MLP on MNIST. We say a barrier is crossed during an update step if there exists a point interpolated between the parameters before and after an update which has a loss value higher than the loss at either points. For most parts of the training, we find that SGD does not cross any significant number of barriers.

The same qualitative analysis for SGD with different hyper-parameters are also shown in section 1 in appendix. The observations we described here remain consistent for all these experiments.

So far we showed qualitative visualizations to make the claim that SGD rarely crosses barriers. In order to show that the claim extends to the rest of the training instead of only a few iterations we showed above, we now quantitatively measure how many barriers are crossed for the entire epoch in different phase of training. This result is shown in table 1 for VGG-11 and Resnet-56 trained on CIFAR-10 (trained for 100 epochs) and an MLP trained on MNIST (trained for 40 epochs). We note that each case, an epoch consists of more than 450 iterations. As we see, a negligible number of barriers are crossed for most parts of the training compared to the number of iterations performed during each epoch. For concreteness, we further compute the number of barriers crossed for the first 40 epochs for VGG-11 on CIFAR-10 as shown in Figure 4 and reach the same conclusion.

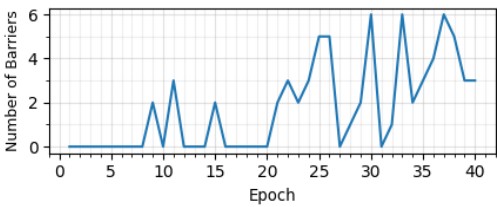

Figure 4: Numbers of barriers found during training loss interpolation for every epoch (450 iterations) for VGG-11 on CIFAR-10. We say a barrier exists during a training update step if there exists a point between the parameters before and after an update which has a loss value higher than the loss at either points. Note that even for these barriers, their heights (defined by $\frac{\mathcal{L}(\theta_t) + \mathcal{L}(\theta_{t+1}) - 2\mathcal{L}(\theta_t^{\min})}{2}$) are substantially smaller compared with the value of loss at the corresponding iterations (not mentioned here), meaning they are not significant barriers.

## 4 THE EFFECT OF BATCH-SIZE ON EXPLORATION

### 4.1 ANALYSIS

Hoffer et al. (2017) discuss that SGD training with different batch-sizes leads to different diffusion rates (very large batch sizes being slower). Further, when training for the same number of epochs, a larger batch training performs less number of iterations. Combining these two observations, they reach the conclusion that large batch training makes the diffusion process slow. As empirical evidence, they show that the distance of parameters from initialization evolves logarithmically in the number of iterations.

We now present a complementary optimization perspective to their observation. To continue, we introduce the following notations. Let $p_i(\theta)$ denote the predicted probability output (of the correct class in the classification setting for instance) of a DNN parameterized by $\theta$ for the $i^{th}$ data sample (in total N samples). Then the negative log likelihood loss for the $i^{th}$ sample is given by $\mathcal{L}_i(\theta) = -\log(p_i(\theta))$. The gradient $\mathbf{g}_B(\theta)$ from mini-batch SGD at a parameter value $\theta$ is expressed as, $\mathbf{g}_B(\theta) = \frac{1}{B} \sum_{i \in B} \frac{\partial \mathcal{L}_i(\theta)}{\partial \theta}$, $\bar{\mathbf{g}}(\theta)$ denotes the expected gradient using all training samples, $B$ is the mini-batch size (and we have also overloaded it to mean the mini-batch set) and $\mathbf{C}(\theta)$ is the gradient covariance matrix at $\theta$. Then the relation between the Hessian $\mathbf{H}(\theta)$ and the dataset gradient covariance $\mathbf{C}(\theta)$ for negative log likelihood loss is described by the Gauss-Newton decomposition as follows,

$$\mathbf{H}(\theta) = \mathbf{C}(\theta) + \bar{\mathbf{g}}(\theta)\bar{\mathbf{g}}(\theta)^T + \frac{1}{N} \sum_{i=1}^{N} \frac{\partial \mathcal{L}_i(\theta)}{\partial p_i(\theta)} \cdot \frac{\partial^2 p_i(\theta)}{\partial \mathbf{w}^2} \tag{2}$$

where $\mathbf{H}(\theta)$ is the Hessian of the loss. The derivation can be found in section B of the appendix.

To continue with our argument, we note that it has been discussed by Shwartz-Ziv & Tishby (2017) that early on during training, the mean gradient over the training set is larger in magnitude compared to the variance in gradients. The above argument essentially says that the scale of mean gradient $\bar{\mathbf{g}}(\theta)$ is larger compared with the scale of $\mathbf{C}(\theta)$. Ignoring the second order term in the Gauss-Newton decomposition above, we see that the mean gradient must be aligned with the top eigenvectors of the Hessian since the scale of gradient covariance is much smaller early on during training. Finally, we note that using large batch-sizes makes the mini-batch gradient closer to the mean gradient by reducing the scale of mini-batch gradient covariance as shown by Hoffer et al. (2017),

$$\text{cov}(\mathbf{g}_B(\theta), \mathbf{g}_B(\theta)) = \left( \frac{1}{B} - \frac{1}{N} \right) \mathbf{C}(\theta) \tag{3}$$

The two arguments together imply that gradients from larger batch-sizes are likely to be more aligned with the high curvature directions of the loss surface especially early on during training.

In convex optimization theory, when gradients point along the top eigenvectors of the Hessian (also referred to as the *sharp directions* of the loss surface), optimization exhibits under-damped convergence, meaning it oscillates along the sharp directions in the case when the learning rate is smaller than a certain threshold. Applying this idea to non-convex loss landscapes, a large alignment between the mini-batch gradient and the sharp directions should also lead to oscillations. At this point, we depart from the conclusions of the convex setting and recall our observation in the previous section that the interpolation between consecutive iterations has a quadratic like shape and SGD moves over barriers for the deep network loss surface. We thus hypothesize that a lower alignment between mini-batch gradients and the sharp directions of the loss surface makes SGD exploration faster by exhibiting less oscillation, and vice-versa.

### 4.2 EMPIRICAL VERIFICATION

Based on the theoretical analysis above, we first conduct experiments to empirically verify that the alignment of mini-batch gradient $\mathbf{g}_B(\theta)$ and hessian $\mathbf{H}(\theta)$ increases when we increase mini-batch size. To do so, we calculate the alignment of mini-batch gradient $\mathbf{g}_B(\theta)$ and hessian $\mathbf{H}(\theta)$ as

$$\frac{\mathbf{g}_B^T(\theta)\mathbf{H}(\theta)\mathbf{g}_B(\theta)}{\|\mathbf{g}_B(\theta)\|_2^2}. \tag{4}$$

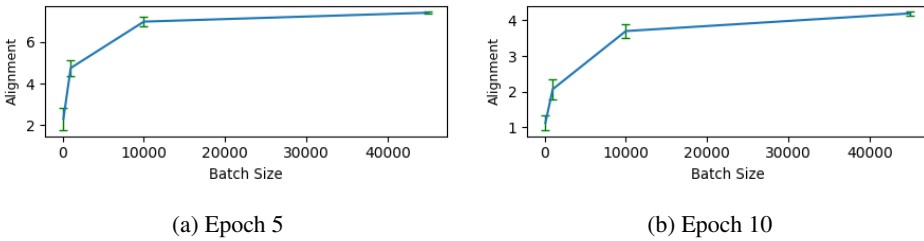

(a) Epoch 5          (b) Epoch 10

Figure 5: Plots for the alignments between mini-batch gradient and hessian with VGG-11 architecture trained using SGD on CIFAR-10 at the end of Epoch 5 and Epoch 10. Alignments are calculated for mini-batch size 100,1000,10000 and 45000 (dataset size).

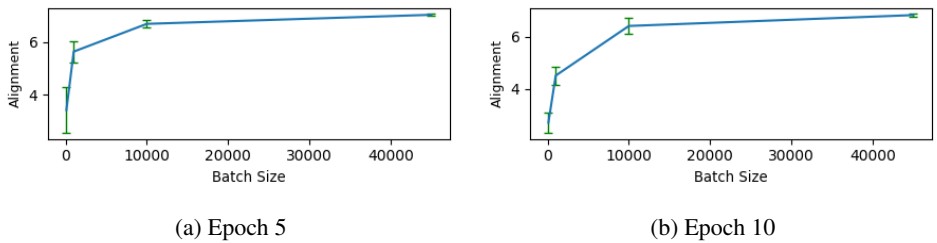

(a) Epoch 5          (b) Epoch 10

Figure 6: Plots for the alignments between mini-batch gradient and hessian with Resnet-56 architecture trained using SGD on CIFAR-100 at the end of Epoch 5 and Epoch 10. All the descriptions are same as described in figure 5.

Figure 5 and figure 6 show the alignments calculated according to Equation 4 on both VGG-11 with CIFAR-10 and Resnet-56 with CIFAR-100 separately at the end of Epoch 5 and Epoch 10. We calculate the alignment for mini-batch size 100, 1000, 10000 and 45000 (which is the dataset size). For every mini-batch size, we sample 50 different batches and calculate the alignment of the current mini-batch gradient with the hessian and show both the mean and standard deviation of alignments in the plots. From both figure 5 and figure 6, we can see that the alignment between mini-batch gradient and hessian is larger for larger mini-batch size.

Based on the empirical verification between mini-batch gradient and sharp directions above, we now verify our argument whether it leads SGD to oscillate in the proximity of the parameter initialization, thus slowing down exploration. Note the latter has been shown by Hoffer et al. (2017). Therefore, to substantiate our claim, we show the degree of oscillation in SGD increases with large batch-size. Specifically, while training deep networks, we keep track of the cosine of the angle between mini-batch gradients from every two consecutive SGD iterations,

$$\cos(\mathbf{g}_{t-1}, \mathbf{g}_t) := \frac{\mathbf{g}_{t-1}^T \mathbf{g}_t}{(\|\mathbf{g}_{t-1}\|_2 \|\mathbf{g}_t\|_2)}. \tag{5}$$

Figure 7 shows the consine calculated according to Equation 5 for Resnet-56 on CIFAR-10 and WResnet on CIFAR-100. Experiments are run with the same learning rate for batch size 500, 5000 and 45000 (dataset size). We can see from the plot that the cosine of the angle between mini-batch gradients from two consecutive iterations remains smaller for larger batch sizes, which indicates that the gradients from two consecutive iterations point more in opposite directions for larger batch sizes. Together with the parameter distance results from Hoffer et al. (2017) that shows that within the same number of iterations, the parameter norm for larger batch sizes is smaller, our experiment verifies that for larger batch sizes, SGD oscillates more in the proximity of the parameter initialization instead of exploring farther away regions.

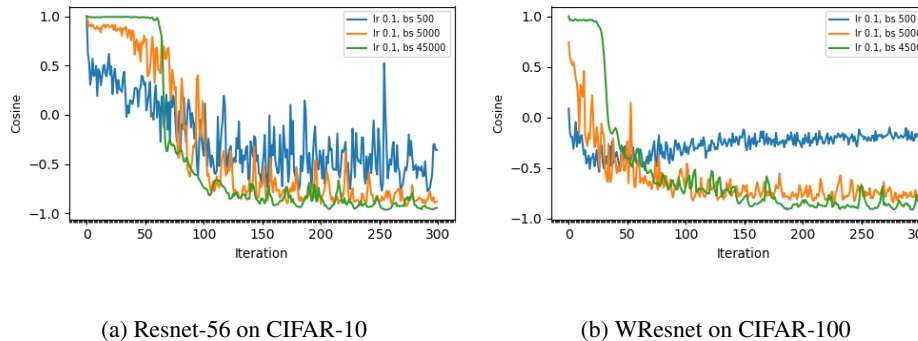

(a) Resnet-56 on CIFAR-10          (b) WResnet on CIFAR-100

Figure 7: Plots of the consine calculated according to Equation 5 for Resnet-56 on CIFAR-10 and WResnet on CIFAR-100 for different batch sizes.

## 5  BACKGROUND AND RELATED WORK

There have been previous work on visualizing the loss surface although from different motivations. Perhaps Goodfellow et al. (2014) is most similar to our work since we use the loss interpolation tool suggested in their paper to perform our analysis. They perform interpolation between the initial and final parameters and based on their finding, draw the conclusion that the loss along the line connecting these two points does not have any barriers. We note that we use their tool for a different purpose and our conclusions are fundamentally different from theirs because we use the observations to investigate whether SGD crosses barriers during optimization over deep networks' loss landscape. Li et al. (2017b) also visualize the loss landscape of different network architectures.

Our work is closely related to a number of recent papers that study SGD as a diffusion process because we present a complementary qualitative view to an aspect of their theory. Hoffer et al. (2017) hypothesize this view based on the evidence that the parameter distance moved by SGD from initialization as a function of the number of iterations resembles a diffusion process. Li et al. (2017a) hypothesize this behavior of SGD and theoretically show that this diffusion process would allow SGD to escape sharp local minima. The authors use this theoretical result to support the findings of Keskar et al. (2016) who find that SGD with small mini-batch size find wider minima. Kushner & Yin (2003); Mandt et al. (2017); Chaudhari & Soatto (2017); Smith & Le (2017); Jastrzebski et al. (2017); Li et al. (2015) study SGD as a discrete approximation of stochastic differential equation under the assumption of a reasonably small learning rate and batch-size (compared with dataset size). Broadly, these papers show that the stochastic fluctuation in the stochastic differential equation simulated by SGD is governed by the ratio of learning rate to batch size. In this paper we study the qualitative roles of batch size without any assumption on how large it is with respect to dataset size.

We note that Zhu et al. (2018) present an analysis of how the structure of gradient covariance matrix can help SGD escape sharp minima more efficiently. Specifically, they show that when the top eigenvectors of the gradient covariance and Hessian are aligned, the escaping efficiency of SGD out of sharp minima is best. We find our view of exploration of different regions by SGD on the other side of the spectrum. While they compare the alignment between the gradient covariance (noise) and Hessian, we talk about the alignment between the mean gradient and the Hessian, which may be seen as complementary views.

There is a long list of work towards understanding the loss surface geometry of DNNs from a theoretical standpoint which is similar in spirit to our analysis of loss surface. Dotsenko (1995); Amit et al. (1985); Choromanska et al. (2015) show that under certain assumptions, the DNN loss landscape is similar to the spherical spin glass model which is well studied in terms of its critical points. Safran & Shamir (2016) show that under certain mild assumptions, the initialization is likely to be such that there exists a continuous monotonically decreasing path from the initial point to the global minimum. Freeman & Bruna (2016) theoretically show that for DNNs with rectified linear units (ReLU), the level sets of the loss surface become more connected as network over-parametrization increases. This has also been justified by Sagun et al. (2017) who show that the Hessian of deep ReLU networks is degenerate when the network is over-parametrized and hence the loss surface

is flat along such degenerate directions. *Broadly these studies analyze DNN loss surfaces (either theoretically or empirically) in isolation from the optimization dynamics.*

In our work we do not study the loss surface in isolation, but rather analyze it through the lens of SGD. In other words, we study the DNN loss surface along the trajectory of SGD, based on which we make deductions about how SGD explores the different regions of the loss surface and the effect of batch-size on this aspect.

## 6    DISCUSSION AND CONCLUSION

Through qualitative results that showed how SGD interacts with the DNN loss surface, we showed evidence that SGD rarely crosses barriers during training. We presented an alternate mechanism that SGD uses to explore different regions of the deep network loss landscape.

We draw similarities between the optimization trajectory in DNNs that we have empirically found, with those in quadratic loss optimization (see section 5 of LeCun et al. (1998)). Based on our empirical evidence, we found that the loss interpolation between parameters from consecutive updates is a quadratic-like shape. This is reminiscent of optimization in a quadratic loss setting with a non-isotropic positive semi-definite Hessian, where the optimal learning rate $\eta$ causes under-damping without divergence along eigenvectors of the Hessian which have eigenvalues $\lambda_i$ such that $\lambda_i^{-1} < \eta < 2\lambda_i^{-1}$.

In the second part of our analysis, we investigated the role of batch-size in exploration, for different regions during SGD optimization of a DNN loss surface. We presented an argument showing mini-batch gradients from larger batch-sizes should align more with the high curvature directions of the loss surface, especially early during training when the scale of mean gradients dominates over gradient covariance. Additionally, we present a complementary view of the exploration aspect of SGD that stems from its diffusion perspective, and show that the alignment of the mini-batch gradient with the sharp directions of the Hessian leads to oscillations preventing SGD from exploring regions far from the initialized parameters.

Finally, much of what we have discussed is based on the loss landscape of specific datasets and architectures along with network parameterization choices like rectified linear activation units (ReLUs) and batch normalization Ioffe & Szegedy (2015). These conclusions may differ depending on these choices. In these cases analysis similar to ours can be performed to see if similar dynamics hold or not. Studying these dynamics may provide more practical guidelines for setting optimization hyperparameters.

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

APPENDIX

## A    OPTIMIZATION TRAJECTORY

This is a continuation of section 3.1 in the main text. Here we show further experiments on other datasets, architectures and hyper-parameter settings. The analysis of GD training for Resnet-56 on CIFAR-10, MLP on MNIST and VGG-11 on tiny ImageNet are shown in figures 8, 16 and 19 respectively. Similarly, the analysis of SGD training for Resnet-56 on CIFAR-10 dataset with batch size of 100 and learning rate 0.1 for epochs 1, 2, 25 and 100 are shown in figures 9, 10, 11 and 12 respectively. The analysis of SGD training for VGG-11 on CIFAR-10 with the batch size of 100 and learning rate 0.1 on epochs 2, 25,100 are shown in figures 13, 14 and 15. The analysis of SGD training for MLP on MNIST for epochs 1 and 2 are shown in figures 17 and 18. The analysis of SGD training for VGG-11 on tiny ImageNet for epochs 1 is shown in figure 20. We also conducted the same experiment and analysis on various batch sizes and learning rates for every architecture. Results of VGG-11 can be found in figures 21, 22, 23 and 24. Results of Resnet-56 can be found in figures 25, 26, 27 and 28. The observations and rules we discovered and described in section 3 are all consistent for all these experiments. Specifically, for the interpolation of SGD for VGG-11 on tiny ImageNet, the valley-like trajectory is weird-looking but even so, according to our quantitative evaluation there is no barrier between any two consecutive iterations.

## B    IMPORTANCE OF SGD NOISE STRUCTURE

Here we derive in detail the relation between the Hessian and gradient covariance for the negative log likelihood loss $\mathcal{L}_i(\theta) = -\log(p_i(\theta))$. Note we use the fact that for this particular loss function, $\frac{\partial \mathcal{L}_i(\theta)}{\partial p_i(\theta)} = -\frac{1}{p_i(\theta)}$, and $\frac{\partial^2 \mathcal{L}_i(\theta)}{\partial p_i(\theta)^2} = \frac{1}{p_i^2(\theta)}$, which yields $\frac{\partial^2 \mathcal{L}_i(\theta)}{\partial p_i(\theta)^2} = \left(\frac{\partial \mathcal{L}_i(\theta)}{\partial p_i(\theta)}\right)^2$.

$$\mathbf{H}(\theta) = \frac{1}{N}\sum_{i=1}^{N}\frac{\partial^2 \mathcal{L}_i(\theta)}{\partial \theta^2} \tag{6}$$

$$= \frac{1}{N}\sum_{i=1}^{N}\frac{\partial}{\partial \theta}\left(\frac{\partial \mathcal{L}_i(\theta)}{\partial p_i(\theta)}\cdot\frac{\partial p_i(\theta)}{\partial \theta}\right) \tag{7}$$

$$= \frac{1}{N}\sum_{i=1}^{N}\frac{\partial^2 \mathcal{L}_i(\theta)}{\partial p_i(\theta)^2}\cdot\frac{\partial p_i(\theta)}{\partial \theta}\frac{\partial p_i(\theta)}{\partial \theta}^T + \frac{\partial \mathcal{L}_i(\theta)}{\partial p_i(\theta)}\cdot\frac{\partial^2 p_i(\theta)}{\partial \theta^2} \tag{8}$$

$$= \frac{1}{N}\sum_{i=1}^{N}\left(\frac{\partial \mathcal{L}_i(\theta)}{\partial p_i(\theta)}\right)^2\cdot\frac{\partial p_i(\theta)}{\partial \theta}\frac{\partial p_i(\theta)}{\partial \theta}^T + \frac{\partial \mathcal{L}_i(\theta)}{\partial p_i(\theta)}\cdot\frac{\partial^2 p_i(\theta)}{\partial \theta^2} \tag{9}$$

$$= \frac{1}{N}\sum_{i=1}^{N}\frac{\partial \mathcal{L}_i(\theta)}{\partial \theta}\frac{\partial \mathcal{L}_i(\theta)}{\partial \theta}^T + \frac{\partial \mathcal{L}_i(\theta)}{\partial p_i(\theta)}\cdot\frac{\partial^2 p_i(\theta)}{\partial \theta^2} \tag{10}$$

$$= \mathbf{C}(\theta) + \bar{\mathbf{g}}(\theta)\bar{\mathbf{g}}(\theta)^T + \frac{1}{N}\sum_{i=1}^{N}\frac{\partial \mathcal{L}_i(\theta)}{\partial p_i(\theta)}\cdot\frac{\partial^2 p_i(\theta)}{\partial \theta^2} \tag{11}$$

where $\bar{\mathbf{g}}(\theta) = \frac{1}{N}\sum_{i=1}^{N}\frac{\partial \mathcal{L}_i(\theta)}{\partial \theta}$.

## C    DISCUSSION

In the main text, we talk about converge in the quadratic setting depending on the value of learning rate relative to the largest eigenvalue of the Hessian. The convergence in this setting has been visualized in **??**.

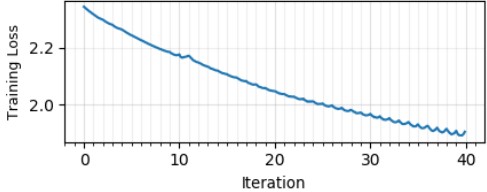

Figure 8: Plots for Resnet-56 Epoch 1 trained using full batch **Gradient Descent (GD)** on CIFAR-10.

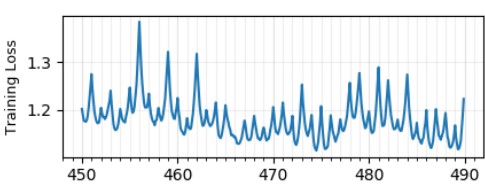

Figure 13: Plots for VGG-11 Epoch 2 trained using **SGD** on CIFAR-10.

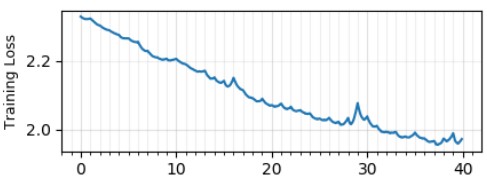

Figure 9: Plots for Resnet-56 Epoch 1 trained using **SGD** on CIFAR-10.

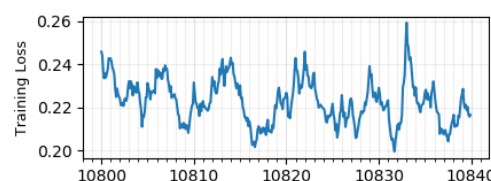

Figure 14: Plots for VGG-11 Epoch 25 trained using **SGD** on CIFAR-10.

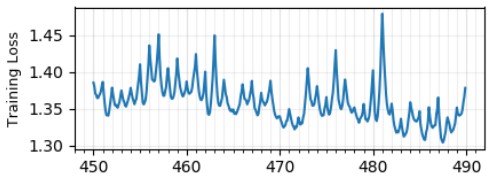

Figure 10: Plots for Resnet-56 Epoch 2 trained using **SGD** on CIFAR-10.

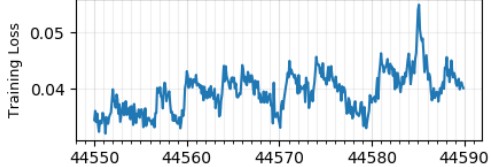

Figure 15: Plots for VGG-11 Epoch 100 trained using **SGD** on CIFAR-10.

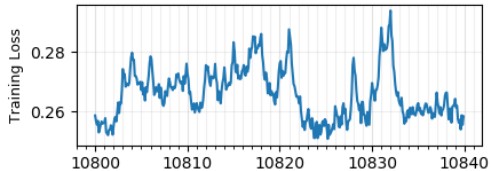

Figure 11: Plots for Resnet-56 Epoch 25 trained using **SGD** on CIFAR-10.

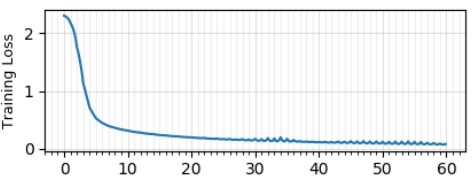

Figure 16: Plots for MLP Epoch 1 trained using full batch **Gradient Descent (GD)** on MNIST.

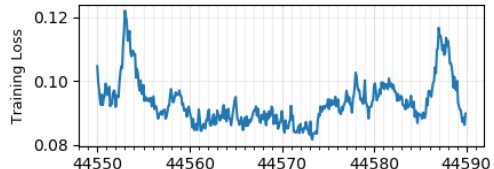

Figure 12: Plots for Resnet-56 Epoch 100 trained using **SGD** on CIFAR-10.

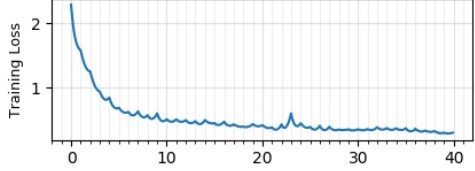

Figure 17: Plots for MLP Epoch 1 trained using **SGD** on MNIST.

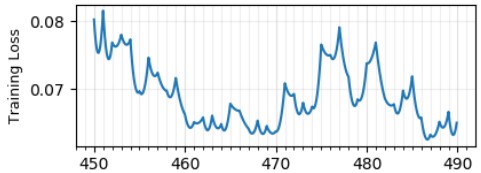

Figure 18: Plots for MLP Epoch 2 trained using **SGD** on MNIST.

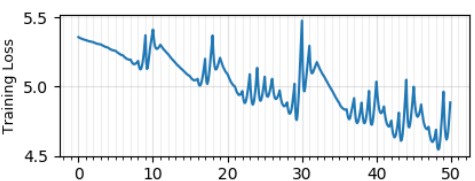

Figure 19: Plots for VGG-11 Epoch 1 trained using full batch **Gradient Descent (GD)** on Tiny-ImageNet.

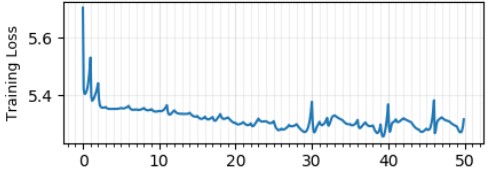

Figure 20: Plots for VGG-11 Epoch 1 trained using **SGD** on Tiny-ImageNet.

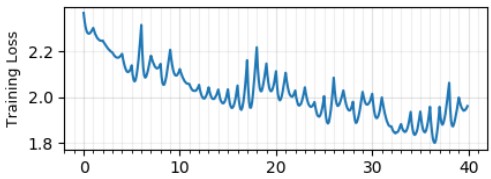

Figure 21: Plots for VGG-11 Epoch 1 trained using learning rate 0.3 batch size 100 on CIFAR-10.

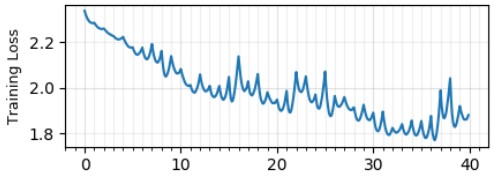

Figure 22: Plots for VGG-11 Epoch 1 trained using learning rate 0.2 batch size 100 on CIFAR-10.

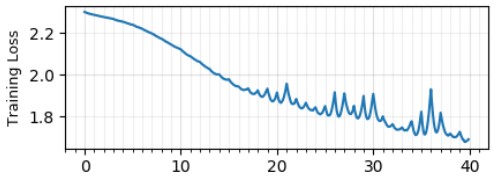

Figure 23: Plots for VGG-11 Epoch 1 trained using learning rate 0.1 batch size 500 on CIFAR-10.

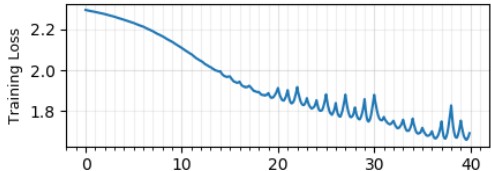

Figure 24: Plots for VGG-11 Epoch 1 trained using learning rate 0.1 batch size 1000 on CIFAR-10.

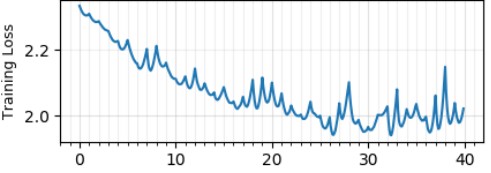

Figure 25: Plots for Resnet-56 Epoch 1 trained using learning rate 0.7 batch size 100 on CIFAR-10.

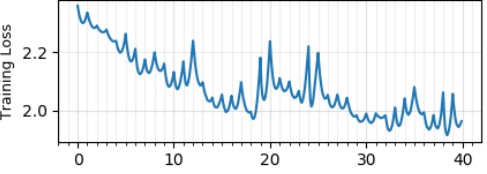

Figure 26: Plots for Resnet-56 Epoch 1 trained using learning rate 1 batch size 100 on CIFAR-10.

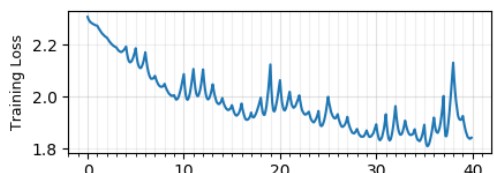

Figure 27: Plots for Resnet-56 Epoch 1 trained using learning rate 1 batch size 500 on CIFAR-10.

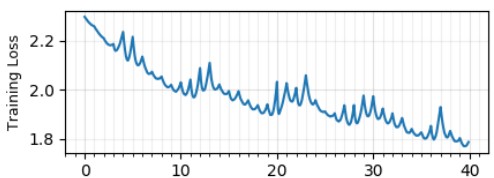

Figure 28: Plots for Resnet-56 Epoch 1 trained using learning rate 1 batch size 1000 on CIFAR-10.

