# OpenReview forum: "A Walk with SGD: How SGD Explores Regions of Deep Network Loss?"
_ICLR.cc/2019/Conference_

### Official Review · AnonReviewer2 · 2018-10-31
**Qualitative analysis lacking key insights and not comprehensive enough?**

**Rating:** 3
**Confidence:** 4

**Review:**

I like the idea of trying to qualitatively illustrate the behavior of SGD when optimizing parameters of complex models, such as Deep and Conv Nets, but I think that the contribution is not very substantial. The connection between SGD and diffusion has been pointed out in previous papers, as acknowledged by the Authors. The study of the effect of batch size is interesting, but again somewhat derived from previous works.

It would helpful to illustrate the difference between "crossing" and "moving over" a barrier with a simple figure.

The experimental validation is interesting, although I think it is limited and perhaps the conclusions that can be drawn from it are not so surprising. I believe it would have been interesting to study other important factors that affect the behavior of SGD, such as learning rate and type of momentum. For example, a larger learning rate might allow for more crossing of barriers. Also, different SGD algorithms (ADAGRAD, ADAM, etc...) would behave considerably differently I expect. At the moment these important factors are overlooked.

It is not clear to me why we would want to avoid larger batch sizes. A larger batch size allows for a lower variance of stochastic gradients, and therefore faster convergence. I think this point requires elaboration, because this forms the motivation behind theoretically grounded and successful SGD works, such as SAGA and the like. I agree that a smaller batch-size is preferable at the beginning of the optimization, but again this is a well known fact (again, see SAGA) and it is for computational reasons mostly (being far away from the (local) mode, a noisy gradient is enough to move in the right direction - no need to spend computations to use an accurate gradient). There is no guarantee that the local optimum close to initialization is a bad local optimum in general, so I don't think that using a large batch size at the beginning is a bad idea for this reason - again it is just computational.

Another thing missing I think is the discussion around why it is potentially a good thing to cross the barrier, either at the beginning of the exploration or towards convergence to a local optimum. At the moment, the paper seems to report the behavior of SGD without key insights on the importance of crossing or avoiding crossing barriers.

As a concluding remark - there has been a lot of work on the connections between diffusions and MCMC algorithms (see e.g., the Metropolis Adjusted Langevin Algorithm - MALA) and a lot of the considerations made in the paper are somewhat known. That is, random walk/diffusion type MCMC (and even gradient-based MCMC like Hybrid Monte Carlo) struggle a lot in non-convex problems and they hardly move across modes of a posterior distribution (equivalent to crossing barriers of potential). So I'm not at all surprised that SGD does not cross barriers during optimization and I would challenge the statement in the introduction saying "Intuitively, when performing random walk on a potential, one would expect barriers being crossed quite often during the process."

---

### Official Review · AnonReviewer3 · 2018-11-01
**Two interesting ideas but somewhat disconnected**

**Rating:** 4
**Confidence:** 3

**Review:**

This paper explores the idea that mini-batch SGD rarely *crosses* barriers during DNN optimization, but rather uses a 'seemingly' or 'alternate' mechanism, as the authors somewhat mysteriously call it on the first page. In the second part of the paper,  they also investigate why the loss surface is explored more slowly when the batch size increases.

I found both parts of the paper reasonably interesting but not too surprising. My main concern is that both parts are , in themselves, not strong enough to warrant publication at ICLR, and the connection between them is rather weak. The authors write 'to complement this finding'  to connect the first to the second investigation, but that's not connecting them very closely is it?
I think it would be better to work out both insights in more detail and publish them in separate papers.
Especially the second insight should be explored more thoroughly. For example, the authors write 'in convex optimization theory, when gradients point along the sharp directions of the loss surface, optimization exhibits under-damped convergence'. This is repeated later in different wordings.  But no reference to this result (I presume it's a mathematical theorem?) is given, neither here nor later when it is said again. The link from the convex to the nonvex DNN case could also be established more convincingly. Everything became quite (too) heuristic at some point...

A few small remarks (which did not influence my judgement):
- while in general (with the exception of the too-fast move from convex to nonconvex that I just explained) the paper is written quite clearly, the prose could be made significantly tighter. For example, the definition of what 'crossing a barrier' means is given three times (!) in the paper (two times in a figure, once in section 2). BTW, isn't it better to say 'moving *around* barriers' rather than 'over' barriers? You now use 'over' but still sounds very similar to just 'crossing'.
- plural nouns are often combined with singular vers ('measurements that ensures'). This happens not just once but all the time...

PROS:
- two nice little ideas; esp. the first one is well-explained
- easy to read
CONS:
- ideas are not very surprising; and just tested on a few data sets; things could be more  robust.
- second idea not fully convincingly explained
- (most important): the two ideas are not closely connected, making this a somewhat strange paper.

---

### Official Review · AnonReviewer1 · 2018-11-05
**Important line of research and novel ideas that lack precision and is limited by mere presentation of observations**

**Rating:** 4
**Confidence:** 5

**Review:**

The subject of how a given algorithm explores the landscape is still a poorly understood area in training neural networks. There is a large body of recent work that attempts to shed light on this puzzle, and each one tries to claim their share in the furthering of the understanding of the relationship between the geometry of the landscape and the dynamics that one chooses in optimization. The present paper is a fine addition to the literature with interesting observations and novel questions, however, it falls short in many core areas: An apparent work in progress that has a great potential.

It is safe to say that "A walk with SGD" has an important single focus in mind: Does the SGD cross over barriers in the weight space of the underlying neural network? This question, at its heart, is intimately linked with the many properties that are attributed to the modern algorithm of the choice and the way it navigates a given non-convex landscape. The paper claims to provide an almost negative answer to the question and thereby busting several myths that are attributed to the "trick" part of SGD algorithm. As good as it sounds, unfortunately, the paper falls short of providing a convincing evidence (be it theoretical or empirical), and the way it tries to frame itself unique and different in relation to related works only indicate a lack of deep understanding of the existing literature. Therefore, I think there are several ways the paper should be improved before it is ready.

A major question (that I hope will easily be addressed) is on the definition of the barrier itself. According to the text, a barrier is defined judging by the minima of two 1-dimensional segments that connect weights connecting three consecutive steps: if the minimum of the line segment defined by the latter step is larger than the former, then it declared that a barrier is crossed. In a low dimensional world, this makes total sense, however, I fail to understand what kind of barrier it implies on the geometry of the landscape: Can the 1-dimensional lines be on the sides of a valley? Can one find *another* 1-dimensional projection for which the inequality is broken? How do such dependencies change the understanding of the problem? And if one is indeed only interested in the flat line segments (since SGD is making discrete steps), then one can, in principle, observe barrier crossing in a convex problem, as well? Is there an argument for otherwise? Or if it is a notion that applies equally well in a convex case then how should we really think about the barrier crossing? On the opposite point of view, can one not imagine a barrier crossing that doesn't appear in this triangular inequality above?

The paper is full of empirical evidence that is guided by a simple observable that is very intuitive, however, it lacks a comprehensive discussion on the new quantity they propose that I consider a major flaw, but that I think (hope) that the authors can fix very easily. Some minor points that would improve the readability and clarity for the reader:
- The figures are not very reader-friendly, this can be improved by better using the whitespaces in the paper but it can also be improved by finding further observables that would summarize the observations instead of showing individual consecutive line interpolations.
- What are the values of the y-axis in Figure 5 and 6? Are they the top eigenvalues of the Hessian?
- In the models that are compared in Figure 7, what are their generalization properties (early stopping and otherwise)?
- The interpretation at the end of p. 6 may be a good motivation for the reader if it had been introduced earlier for that section.

Finally, Section 5 reads very strangely, I have hard times understanding why certain phrases exist the way they are in this part. Here are further notes on Section 5:
- Why the whole first paragraph focused on hot the current paper is different than Goodfellow et al (2014) (it is obvious that it is for a different purpose), or why do we read the sentence "Li et al (2017b) also visualize...." What way do they visualize, is it the only paper that does visualization, what's the relation with the current paper and barrier crossing?
- For the second paragraph, I can suggest another paper, https://arxiv.org/abs/1803.06969, that was at ICML which also looks at the diffusion process through the parameter distance at different times which is similar to Hoffer et al. which also claims no barrier crossing similar to the present paper.
- However, my main issue is the exact connection between diffusion and no barrier crossing and it's connection to SGD preferring wide local minima instead of a narrow one. The second paragraph of the conclusion touches upon this subject. But it is not entirely clear how they are linked (except for the brittle SDE approximation at Li et al (see https://arxiv.org/abs/1810.00004)). Overall, the paper would benefit a lot from the discussion on why it is preferable to have SGD choose one basin over another in the beginning, as it is, it looks like the paper has another agenda behind the scenes.
- In the fourth paragraph of the conclusion, the paper refers to three papers that link DNN to spin glasses, in two of the (older) references the networks are far from what we have today, and the third one is far from "showing" anything between DNN and spin glass. In any case, what's the link between the aspects studied there with the present paper?
- Finally, the paper claims at the last few sentences that the works referred a little bit earlier look at the loss surface "in isolation from the optimization dynamics", however, many of those works cited have their empirical observations much like the current paper, and clearly they all "study the DNN loss surface along the trajectory of SGD" necessarily as it is the way to find local minima, saddle points, paths, curvature etc... The present paper is already very interesting and full of novel insight, I fail to see the value of struggling to stand out like this.

Overall, I think the paper is a very interesting step forward in understanding SGD dynamics on the DNN landscape. And, even though it has many shortcomings as it currently stands, I think it has a lot of room to improve.

---

### Meta-Review · Area_Chair1 · 2018-12-13

**Confidence:** 5
**Recommendation:** Reject

**Metareview:**

The reviewers agree that the paper needs significantly more work to improve presentation and is not fully empirically and conceptually convincing.